# A Reliability Investigation of VDMOS Transistors: Performance and Degradation Caused by Bias Temperature Stress

**DOI:** 10.3390/mi15040503

**Published:** 2024-04-05

**Authors:** Emilija Živanović, Sandra Veljković, Nikola Mitrović, Igor Jovanović, Snežana Djorić-Veljković, Albena Paskaleva, Dencho Spassov, Danijel Danković

**Affiliations:** 1Faculty of Electronic Engineering, University of Niš, Aleksandra Medvedeva 14, 18000 Niš, Serbia; emilija.zivanovic@elfak.ni.ac.rs (E.Ž.); sandra.veljkovic@elfak.ni.ac.rs (S.V.); nikola.i.mitrovic@elfak.ni.ac.rs (N.M.); igor.jovanovic@elfak.ni.ac.rs (I.J.); 2Faculty of Civil Engineering and Architecture, University of Niš, Aleksandra Medvedeva 14, 18000 Niš, Serbia; snezana.djoric.veljkovic@elfak.ni.ac.rs; 3Institute of Solid State Physics, Bulgarian Academy of Sciences, Tzarigradsko Chaussee 72, 1734 Sofia, Bulgaria; paskaleva@issp.bas.bg (A.P.); d.spassov@issp.bas.bg (D.S.)

**Keywords:** VDMOS, reliability, NBTI, responsible mechanisms, self-heating

## Abstract

This study aimed to comprehensively understand the performance and degradation of both p- and n-channel vertical double diffused MOS (VDMOS) transistors under bias temperature stress. Conducted experimental investigations involved various stress conditions and annealing processes to analyze the impacts of BT stress on the formation of oxide trapped charge and interface traps, leading to threshold voltage shifts. Findings revealed meaningful threshold voltage shifts in both PMOS and NMOS devices due to stresses, and the subsequent annealing process was analyzed in detail. The study also examined the influence of stress history on self-heating behavior under real operating conditions. Additionally, the study elucidated the complex correlation between stress-induced degradation and device reliability. The insights contribute to optimizing the performance and permanence of VDMOS transistors in practical applications, advancing semiconductor technology. This study underscored the importance of considering stress-induced effects on device reliability and performance in the design and application of VDMOS transistors.

## 1. Introduction

In the domain of advanced CMOS devices, negative bas temperature instability (NBTI) emerges as a prominent reliability concern. Typically observed in p-channel metal oxide semiconductor (PMOS) transistors operating at elevated temperatures (ranging from 100 to 250 °C) under negative gate oxide fields spanning 2 to 6 MV/cm, NBT stress-induced threshold voltage shifts have acquired considerable attention. On the contrary, n-channel metal oxide semiconductor (NMOS) transistors have often been observed as less susceptible to NBTI-related effects due to their limited exposure to negative gate bias during operation. However, investigations unveiled a potential vulnerability in NMOS transistors subjected to high negative gate bias in some scenarios, leading to significant instabilities induced by NBT stress [1,2,3,4].

NMOS transistors are very useful components with a range of applications, starting from the fundamental tasks of switching and amplification to other important applications like motor control, battery management, voltage regulation, power management, and lighting systems. Their usefulness extends further into specialized domains such as automotive electronics, high-frequency applications, and renewable energy systems. With their capability to efficiently handle high currents and voltages, NMOS transistors are integral to the fabric of modern electronic circuits, enabling innovation and progress across a wide spectrum of industries and technologies.

In applications where the NMOS transistor is occasionally utilized under negative bias, as previously mentioned, there exist scenarios where accelerated transistor turn-off is important. To enable its operation, a gate voltage above the threshold voltage is initially applied. For instance, if the *V*_T_ of an NMOS transistor is 3.6 V, a gate voltage higher than the threshold voltage, such as 10 V, is applied to ensure channel opening. When the transistor needs to be turned off, the gate voltage is then reduced to 0 V. Since the transistor channel remains closed at this point, the transistor is effectively turned off. However, in some cases, there is a need for immediate transistor termination due to specific application requirements. This is achieved by introducing a negative voltage, such as −10 V, beyond the threshold voltage. While this ensures rapid transistor turn-off, it also gives rise to mechanisms characteristic for NBTI as an unintended consequence.

It is important to note that in some applications, MOS devices can be used for gas sensors, photoconductors, transistor-based electronics, and especially for neuromorphic transistors [5,6]. Over the past few years, field-effect transistors (FET) have experienced expansion, and an increasing number of authors are conducting research on their reliability. There has been significant research conducted both domestically and internationally on electrospinning technology and its applications in the field of FETs based on metal oxide nanowires [7,8,9,10]. Modern materials are used for both NMOS and PMOS transistors, and nanowire MOSFETs are being developed for various applications, which has spurred the realization of numerous experiments and the characterization of these components.

Power metal oxide semiconductor field effect transistors (MOSFETs) are frequently subjected to high levels of current and voltage during their operation, leading to challenges associated with self-heating. The significance of NBTI goes beyond theoretical domains, particularly in the context of power MOSFETs. Previous research efforts have been concerned with elucidating degradation mechanisms linked to NBTI and estimating the operational permanency of p-channel power vertical double diffused MOSFETs (VDMOSFETs) [11]. Expanding upon this groundwork, the current study undertook a comprehensive analysis of the bias temperature (BT) stressing (Table 1) and subsequent low gate bias annealing on both p- and n-channel VDMOSFETs [11].

Using commercially available p-channel (IRF9520) and n-channel (IRF510) power VDMOSFETs made using standard Si-gate technology with a gate oxide thickness of about 100 nm, the investigation aimed to unveil insights into the performance dynamics of these devices. According to the technical documentation provided by the manufacturer, the components are designed to withstand currents of up to 6.8 A and drain-source voltages of −100 V for p-channel devices. As for the n-channel devices, they exhibit comparable characteristics, with maximum current ratings of about 5.6 A and drain-source voltages of +100 V [13,14]. Both components are encapsulated in TO-220 packages, which include space for a heatsink mounting. The heatsink was installed during the experimental procedure.

Throughout the years, the investigation of NBTI has attracted significant academic attention, as proven by the group of authors who have contributed to this field [15,16,17,18,19,20,21]. NBTI is investigated across a spectrum of semiconductor components, including FinFETs, integrated circuits (ICs), where the effect can impact the reliability and performance of digital and analog circuits. Of particular relevance to this review were power VDMOS transistors [16,17,18]. Several experimental methods have been developed to obtain appropriate experimental data [15,21]. However, the mechanisms that fully explain these experimental data have not yet been thoroughly elucidated and remain the subject of investigation [19,22]. Furthermore, various models in the form of different equivalent circuits have been developed [20,21]. An increasing number of researchers are engaged in studying NBTI, and our research team has been involved in these studies for more than two decades. The initial research phase of this team was an investigation that involved examining the mechanisms responsible for the observed effects in PMOS transistors subjected to static NBT stress. Subsequently, reliable operation models were considered, and enhanced experimental setups were developed. The examination then expanded to encompass pulsed NBT stress and to include NMOS transistors. Based on numerous experimentally obtained results, various electrical circuits were created to model the observed threshold voltage shifts. Finally, during the experimental process, a combination of NBT stress with other forms of stresses such as radiation, magnetic fields, and self-heating were applied. These investigations are currently ongoing, see references therein [12].

## 2. Experimental Method

In order to conduct a comprehensive analysis of the bias temperature stressing effects in NMOS and PMOS transistors, investigated VDMOS transistors were subjected to various stress conditions. It is worth noting that the experiment was organized such that stressing, and transistor characteristic measurements were alternating, according to predefined time intervals. Additionally, attention was taken during the experiment to handle the components to avoid electrostatic discharge and to ensure that the components did not lose functionality before the end of the experiment.

The BT stress application began first and Figure 1 represents the diagram of the experimental setup for component BT stress. Applied gate voltage was either static (*V*_G_ = −40 V) or pulsed (*V*_G_ = −40 V, *f* = 10 kHz and DTC = 50%). Both the signal generator and power supply were required in order to effectively obtain the gate voltage that was required. Placing the oscilloscope at the input of the circuit allowed it to monitor the pulsed signal that was being transmitted to the component under test. The use of the heating chamber was necessary to achieve a high temperature during BT stress and annealing. The temperatures of 125 °C, 150 °C, and 175 °C were established via heating chamber.

In Figure 1, the PMOS transistor denoted as S1 was used as a switch to enable the pulsed signal to reach the gate of the investigated set of components, marked as T1. The tested set of PMOS transistors, marked as T2, was under static stress. Both sets of components, T1 and T2, were placed in a heating chamber at 150 °C for the whole BT stress period. The source enabled realization of both negative bias and positive bias stress; the applied negative voltage is shown in Figure 1.

For the purpose of determining the *I*-*V* transfer characteristics during BT stress, the SMU (source measurement unit) Keysight B2901A (Santa Rosa, CA, USA) system was utilized [23]. Figure 2 depicts the arrangements that were made for the measurement. Using the computer made it simpler to maintain control over the SMU. The stress voltage, which was static or pulsed, had to be removed from the device that was being tested in order to carry out *I*-*V* measurements using this method. After that, the threshold voltage was determined based on the transfer characteristic that has been measured.

## 3. Current-Voltage Characteristics

In Figure 3 are presented the NMOS and PMOS transfer characteristics shifts caused by the NBT stress of 168 h. For NMOS transistors, the *I*-*V* characteristics shifted to the left, indicating a decrease in the threshold voltage, i.e., *V*_T_ decreases. For PMOS transistors, the *I*-*V* characteristics also shifted to the left, again showing a decrease in the threshold voltage. However, if we consider the absolute value for PMOS devices, it indicated an increase. For illustration purposes, the characteristics shown here were recorded when the components were tested under negative gate voltage, specifically at −40 V. Only the initial and final characteristics are shown in the figure, while the intermediate characteristics were omitted for clarity of the graph. The number of measured characteristics can be seen in Figure 4. Based on the measured experimental characteristics, changes in the threshold voltage were determined and are displayed in Figure 4.

Regarding the fact that both NMOS and PMOS transistors were stressed, the results were obtained under both positive and negative gate voltages of 40 V, as presented in Figure 4. The changes obtained via applying negative voltage are denoted as NBTI, while changes obtained by applying positive voltage are denoted as PBTI (positive bias temperature instabilities). Moreover, from previous research and the literature [22,24,25,26], it is known that larger changes occur at higher voltages, hence the results at −45 V are shown in this figure, too. An experiment involving stress at +45 V was not of interest. As we can see from this figure, the results at +40 V were negligible for both NMOS and PMOS transistors. In Figure 4, it is illustrated that the NBT stress under typical conditions yielded practically identical *V*_T_ shifts in p- and n-channel VDMOS devices, which is a characteristic unique only to these components [6]. The corresponding PBT stress, however, did not seem to significantly affect threshold voltage in either of the two device types.

Previous studies have extensively analyzed and explained the change in Δ*V*_T_ according to the *t*^n^ law [27]. Only two phases are visible in this figure because the experiment lasted for 168 h; for the third phase to be visible, the experiment needed to last much longer [27].

It was necessary to fully analyze the parameter *n* (in power law) in a continuous manner over the stress regime; the results of the detailed study are explained in Figure 5. This graph represents the results obtained at a gate voltage of −45 V, including observations made under both static and pulsed stress conditions. The thorough examination of outcomes in the presence of dynamic stress conditions was of great importance, as it reflected the dynamic operating conditions experienced in real applications. Researchers could simulate the variations and periodic load conditions that occur during normal operation by subjecting the devices to pulsed stress. On the basis of both static and pulsed stress, vital insights can be provided into device reliability under real situations.

Moreover, it is crucial to emphasize that these experiments were carefully established and carried out at different temperatures. To fully evaluate the durability and operational consistency of the devices in different conditions scenarios, the experiments were carried out under various thermal conditions, taking into account the significant impact of temperature on device performance and reliability. This comprehensive approach guaranteed that the results were both accurate and representative for the various operational environments found in real operation modes, hence increasing the significance and practicality of the study findings.

## 4. Oxide Trapped Charge and Interface Traps

Observed threshold voltage shifts in the tested samples were caused by oxide trapped charge (*N*_ot_) and interface traps (*N*_it_) created during BT stress applied to power VDMOS transistors. Most of the literature data accentuate that NBTI can be of importance only in p-channel MOSFETs [28]; it was studied for several decades. It is well known that the threshold voltage shifts due to stress-induced oxide trapped charge and interface traps in PMOS and NMOS transistors can be expressed, respectively, as [29]:(1)$$\Delta V_{Tp}=-\frac{q\Delta N_{otp}}{C_{ox}}-\frac{q\Delta N_{itp}}{C_{ox}},$$
(2)$$\Delta V_{Tn}=-\frac{q\Delta N_{otn}}{C_{ox}}+\frac{q\Delta N_{itn}}{C_{ox}},$$
where *q* is the elementary charge and *C_ox_* is the gate oxide capacitance per unit area. Assuming that the NBT stress creates similar amounts of oxide trapped charge and interface traps in both PMOS and NMOS devices, the net effect on threshold voltage, Δ*V*_T_, must be greater in p-channel devices, as only in this case, the positive oxide charge and positive interface charge were superimposed. Moreover, the NMOS devices were not operating under the negative gate bias, so the NBTI generally was not considered of importance in NMOS devices. However, having in mind that high negative gate bias can be used in some automotive applications for the faster turning off of the NMOS devices, rather significant NBT stress-induced threshold voltage shifts have been found in n-type trench DMOS transistors [30]. Previous research of NBTI in power VDMOSFETs led to a rather similar finding [31,32].

The dependencies of the underlying buildup of gate oxide trapped charge and interface traps on stress time under stress bias (at 150 °C) and temperature (at −/+40 V) are illustrated in Figure 6, Figure 7 and Figure 8. The subthreshold midgap technique (SMGT) was used for assessing Δ*N_ot_* [33], while Δ*N_it_* was evaluated using both SMGT and the charge pumping technique (CPT) [34].

As demonstrated, only stress biases with negative gate polarization induced significant threshold voltage changes in both p- and n-channel VDMOS power transistors. However, due to the varying impact of charge density in the oxide and interface states on threshold voltage changes in p- and n-channel MOS transistors, a comprehensive qualitative and quantitative analysis was necessary. Therefore, Figure 6 and Figure 7 depict the time dependence of Δ*N_ot_* and Δ*N_it_* during NBT and PBT stresses in both p- and n-channel VDMOS power transistors, determined using the SMG technique [33].

Observations gathered from Figure 6 and Figure 7, presented at the same scale for easier comparison, yielded the following insights. In both types of transistors, the oxide trapped charge during NBT stress was substantially greater than that during PBT stress. Firstly, Δ*N_ot_* in NMOS transistors during both NBT and PBT stresses was significantly higher than in PMOS transistors during corresponding stresses. Δ*N_it_* exhibited qualitatively similar trends to Δ*N_ot_* in both types of transistors and during both NBT and PBT stresses. It can be seen that Δ*N_ot_* formed during NBT stress surpassed Δ*N_it_*. However, two notable differences were evident. Namely, Δ*N_it_* initially increased more rapidly than Δ*N_ot_* during NBT stress but slowed down in the second phase, entering saturation much faster. Δ*N_ot_* formed during PBT stress approached similar values as Δ*N_it_*, with Δ*N_ot_* initially higher but converging by the end of stress. In NMOS transistors, Δ*N_it_* even surpassed Δ*N_ot_* during the PBT stress.

It is known that SMG technique measurements essentially operate in a static regime, yielding interface state densities that can be significantly higher than the actual values of real or so-called fast interface states. Given that charge exchange with the substrate during static or low-frequency measurements can occur through border trap centers, this technique registered these centers as interface states [35,36]. Considering the earlier observation that NBT stress in p-channel VDMOS power transistors did not lead to significant formation of border trap centers, it begged the question of whether this held true during PBT stress and what the situation was with n-channel samples during voltage-temperature stresses. Thus, for the determination of changes in interface trap density in n-channel samples, additionally, the CP technique was used, based on measurements at a frequency of 100 kHz, which has been shown to enable precise, high-resolution, and direct measurements of interface trap density in stressed samples [34]. Therefore, Figure 8 depicts the temporal dependencies of Δ*N_it_* during NBT and PBT stresses in both p- and n-channel VDMOS power transistors, determined using the CP technique [34].

Through detailed analysis of the results presented in Figure 7 and Figure 8, shown at the same scale for comparison, the following important conclusions can be drawn. Firstly, Δ*N_it_* determined via the CP technique in both sample types was almost identical under identical voltage-temperature stress conditions. Secondly, Δ*N_it_* determined via the CP technique exhibited significantly higher values in NBT-stressed samples compared to PBT-stressed samples (by almost an order of magnitude). Changes in interface trap density obtained via SMG and CP techniques were approximately equal in p-channel samples during both PBT and NBT stresses, indicating that significant formation of border trap centers did not occur in p-channel VDMOS power transistors not only during NBT, but also during PBT stress. Changes in interface trap density obtained via the SMG technique were much larger than those obtained via the CP technique in n-channel samples, both during PBT and NBT stresses, indicating that significant formation of border trap centers occurred in n-channel VDMOS power transistors during voltage-temperature stresses.

## 5. NBTI Mechanisms

It is well known that the BTI are the result of a buildup of oxide trapped charge and interface traps due to stress-initiated electrochemical processes involving oxide and interface defects, holes, and a variety of species (H^•^, H_2_, H^+^, OH, H_2_O, and H_3_O^+^). As already mentioned, regardless of decades of studies, the responsible mechanisms are not clearly established [20,37,38].

During NBT stress, under the influence of the electric field, positive charge rapidly accumulates in the oxide by capturing available holes at oxygen vacancies distributed near the silicon-silicon dioxide interface [39]:(3)$$O_{3}\equiv Si^{•\;•}Si\equiv O_{3}+h^{+}\to O_{3}\equiv Si^{+\;•}Si\equiv O_{3}.$$

Furthermore, the formation of the positive charge in the oxide can also occur through the breaking of weak ≡Si_O_-H bonds near the silicon-silicon dioxide interface facilitated by available holes [40], as presented in Figure 9:(4)$$O_{3}\;\equiv \;Si\;-\;H+h^{+}↔O_{3}\;\equiv \;Si^{+}+H^{•}.$$

The accumulated positive charge in the oxide leads to a reduction in the local field near the silicon-silicon dioxide interface, thereby diminishing the rate of oxide trapped charge formation during the subsequent stages of NBT stressing, with a possibility of its transformation into interface traps. However, these processes alone cannot fully account for the rapid increase in *N*_it_ in the initial phase of stress, primarily due to the shortage of electrons required for their formation. Thus, it seems likely that the dissociation process of the weakest ≡Si_Si_-H bonds at the interface under the influence of a strong electric field also plays a significant role during this phase [41]:(5)$$Si_{3}\equiv Si-H↔Si_{3}\equiv Si^{•}+H^{•}.$$

Since the neutral hydrogen atom (H^•^) released during the dissociation process is highly reactive, it can also participate in the dissociation of ≡Si-H bonds (at the interface or in the oxide near the interface), leading to the formation of new interface traps or a positive charge in the oxide. However, due to its strong reactivity and instability, it tends to dimerize into hydrogen molecules (H_2_) or react with holes to form hydrogen ions (H^+^) [42]:(6)$$H^{•}+H^{•}\to H_{2},$$
(7)$$H^{•}+h^{+}\to H^{+}.$$

## 6. Annealing of p-Channel and n-Channel Power VDMOSFETs

In order to gain a deeper understanding of NBTI-related phenomena in n- and p-channel devices, the transistors were subjected to additional stress. Both the IRF9520 (p-channel) and IRF510 (n-channel) devices were exposed to NBT stressing and subsequently gate bias annealing was conducted. During the experiment, three negative gate voltages were applied to three distinct temperatures for the purpose of stressing. The focus was mainly on the standard results obtained from the subsets of PMOS and NMOS devices that were strained with −40 V at 150 °C. Both subgroups were initially subjected to stress for the duration of one week, which has previously been determined to be enough time for the stress-induced *V_T_* shift to reach the saturation phase [27,39,43]. Following the application of stress, each subset was separated into three groups of devices. These groups were then subjected to one week of annealing at a temperature of 150 °C, with each group being exposed to a different gate bias (−10 V, 0 V, and +10 V). The devices were periodically evaluated using the identical current-voltage (*I*-*V*) measurements during both the stressing and annealing processes.

The voltage threshold shifts recorded in both types of devices under the influence of the above negative bias temperature stress and gate bias anneal sequence are depicted in Figure 10. It was evident that the initial negative bias temperature stress caused voltage threshold alterations of around 0.3 V in both PMOS and NMOS devices. For PMOS devices (as shown in Figure 10), annealing with a negative gate bias did not result in any noticeable alteration to the *V_T_* shift caused by the initial stressing. However, annealing with a zero or positive gate bias reduced the *V_T_* shifts induced by initial stress by approximately 50%. The positive bias annealing demonstrated the most efficiency in restoring the stress-induced *V_T_* shift, while the maximum recovery was only marginally superior to that observed in devices annealed under zero bias. Conversely, for NMOS devices (as shown in Figure 10), annealing had distinct impacts on the threshold voltage under each of the three applied gate bias settings. The restoration of the threshold voltage was minimal when subjected to negative bias, but considerably more pronounced under zero bias. However, when a positive gate bias annealing was applied, the threshold voltage fully recovered within one hour, and subsequently exceeded the initial value of *V_T_* before the initial NBT stress.

Upon revisiting the data presented in Figure 10, it became apparent that the largest deviation of the threshold voltage in p-channel devices was just below 0.3 V. This deviation was observed at the conclusion of the NBT stress. Subsequent annealing under any bias did not amplify this deviation, but rather diminished it. The threshold voltage of NMOS devices had a drop of 0.3 V throughout the stress. Additionally, applying zero or negative gate bias during the annealing process further minimized the shift caused by stress. Nevertheless, subjecting the device to annealing with a positive gate bias, which is the typical operational bias for NMOS devices, resulted in not only a complete recovery of the threshold voltage, but also an increase of approximately 0.15 V over the value before the stress. Consequently, the overall variation observed throughout the entire stress and annealing process reached approximately 0.45 V. This strongly suggested that if the NMOS devices were subjected to a negative gate bias and increased temperature at any point during their operation, the associated instabilities could be more severe compared to those observed in PMOS devices.

## 7. Self-Heating under Real Operating Conditions

It is known that VDMOS devices are primarily used in switching power supplies, automotive, and space industries [34,35,36,37,38,39,40,41,42,43,44,45,46,47]. The majority of these applications rely on the VDMOS devices’ superior switching capabilities, so the devices can function as efficient switches. Therefore, these devices are primarily utilized when the controlling signal has a pulsed waveform. The devices have an operating switching frequency in the MHz range, which makes them suitable for various circuit applications. The controlling signal’s characteristics, such as duty cycle, rise time, and fall time, dictate the transistor’s on-time and off-time. When the controlling signal’s voltage surpasses *V_T_*, the VDMOS acts as a closed switch, otherwise, it functions as an open switch. However, during the operation, the threshold voltage (*V_T_*) of the VDMOSFET changes due to self-heating [48]. Considering all these points, the objective of this experimental segment was to determine how previous stresses affect real operating conditions.

Figure 11 displays the absolute temperature changes over time under real operating conditions. Figure 11a shows the outcomes for PMOS devices, each subjected to distinct pre-stress histories. The initial set involved devices without any prior stresses. The subsequent set entailed using devices previously subjected to pulsed NBT stress, while the last set included devices previously stressed via static NBT stress. In addition to more detailed analysis of the heating, the subsequent cooling during the harshest conditions was also measured. These components exhibited the most significant temperature changes, and they needed the longest cooldown period, which is why these results are shown here.

During heating, all component groups were exposed to a pulsed signal with parameters representing typical signals encountered in switching power supplies: a frequency of 1 Hz, *t_rise_* and *t_fall_* of 100 ms, and a duty cycle of 50%. For obtaining the pulsed signal, the Agilent Waveform generator 33521A (Santa Clara, CA, USA) was used. An active load Rigol 3021 (Beijing, China)was also included in the drain circuit to maintain a constant current of 1 A through the transistor. These parameters were selected based on their common existence in practical applications. A similar setup was used for testing the NMOS devices. Figure 11b illustrates NMOS devices previously exposed to static NBT stress, specifically selected due to the highest temperature fluctuations observed under simulated real operating conditions, making them suitable for comparison with Figure 11a. The temperature in NMOS transistors was higher than in the PMOS transistors. Regarding the fact that PMOS and NMOS transistors have different maximum currents (6.8 A and 5.6 A, respectively), it can be expected that they behave differently with the same current flowing through them.

The graphs in Figure 11 show a temperature dependence that changed over time, but Δ*T* increase did not have the same dynamics. The increase in the chip’s temperature was mainly caused by the effects resulting from power dissipation. Each transition of the pulses created additional stress on the device. Temperature rose during each edge’s duration and decreased towards thermal balance during the off-state. The temperature increased steadily with an increased number of pulses [49]. Additionally, in PMOS transistors, already stressed samples showed changes in threshold voltage, with an increase in the absolute threshold voltage value. This increase extended the duration of channel opening at the same gate voltage, leading to current flowing through higher resistance in stressed devices for longer periods, which led to power dissipation.

Previous research and the existing literature have shown that devices that have been stressed in the past are more likely to experience self-heating compared to new samples [49]. It was also proved that larger changes have been recorded in static than in the pulsed NBT-stressed components. Previous processing of devices led to parameter degradation, as shown by changes in threshold voltage. Changes in the threshold voltage affected the formation of the channel, causing a delay in channel opening. This delay allowed the current to pass through while resistance was increased, leading to higher power dissipation and subsequent self-heating. During the experiment, device pictures were taken during the measurement process. The first image depicts the component during stress, illustrating the gradual change in the temperature over time captured via infrared thermographic camera Varioscan 3021 (Puchheim, Germany) (Figure 12a). The photo of one of the measured devices is depicted in Figure 12b. On the protoboard, there were wires or contacts through which current flowed, or voltages were applied to specific pins throughout different conductors.

## 8. Conclusions

The influence of voltage-temperature stresses on both p- and n-channel power VDMOS transistors was investigated in this study. A notable observation was that negative bias temperature instability affects n-channel power VDMOS transistors, which was not necessarily characteristic of other types of power components. It is worth noting that, despite significant differences in the contribution of oxide trapped charge and interface states, almost identical threshold voltage shifts occurred with both p- and n-channel power VDMOSFETs. Both types of transistors had a much higher oxide trapped charge during NBT stress than PBT stress. NMOS transistors exhibited significantly higher Δ*N_ot_* during NBT and PBT stresses compared to PMOS transistors. Additionally, Δ*N_it_* and Δ*N_ot_* showed comparable trends in both types of transistors and under NBT and PBT stresses. If the NMOS devices are exposed to negative gate bias and higher temperature during operation, it is more likely that the resulting instabilities would be more pronounced than those seen in PMOS devices. 

Additionally, the previous treatment of the tested devices is highly significant. Specifically, components previously exposed to static and pulse gate bias stresses exhibited more pronounced self-heating under normal operating conditions, as confirmed via results obtained from thermographic cameras. Self-heating degrades component parameters and thereby reduces the period of reliable operation of the components. As the mechanisms responsible for changing these parameters are not fully revealed, further extensive research will be carried out. Components were planned to be tested under simultaneous exposure to NBT and other forms of stress such as magnetic fields and radiation. Subsequently, a detailed analysis and comparison of how these conditions affect the self-heating process are intended.

## Figures and Tables

**Figure 1 micromachines-15-00503-f001:**
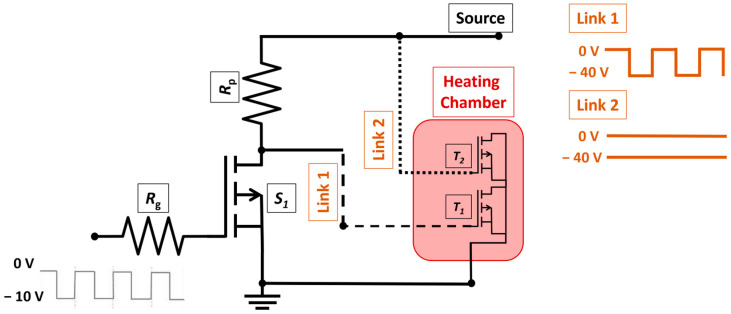
Diagram of the experimental setup for BT stress of investigated components.

**Figure 2 micromachines-15-00503-f002:**
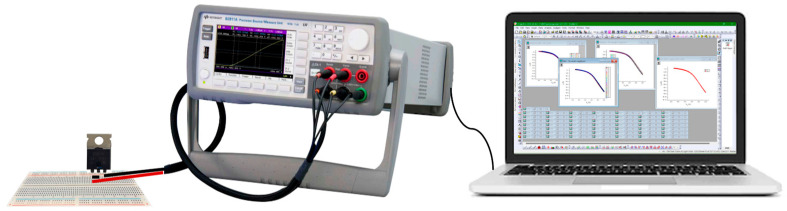
Schematic representation of setup for *I*-*V* transfer characteristics measuring.

**Figure 3 micromachines-15-00503-f003:**
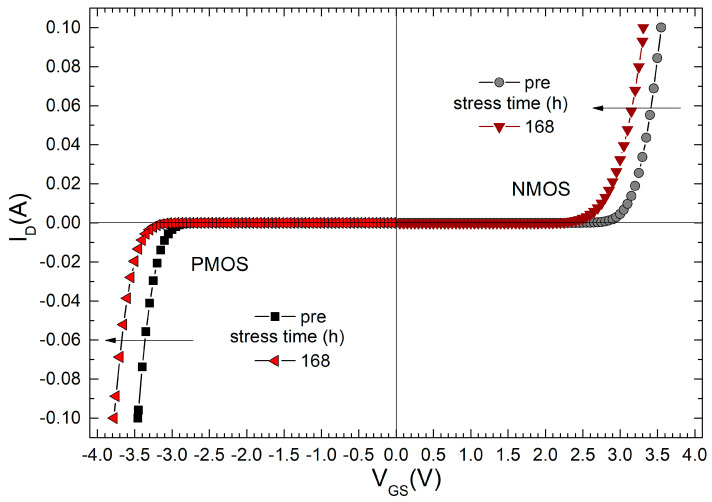
NMOS and PMOS transfer characteristics shifts caused by NBT stress (*V*_G_ = −40 V, *T* = 150 C).

**Figure 4 micromachines-15-00503-f004:**
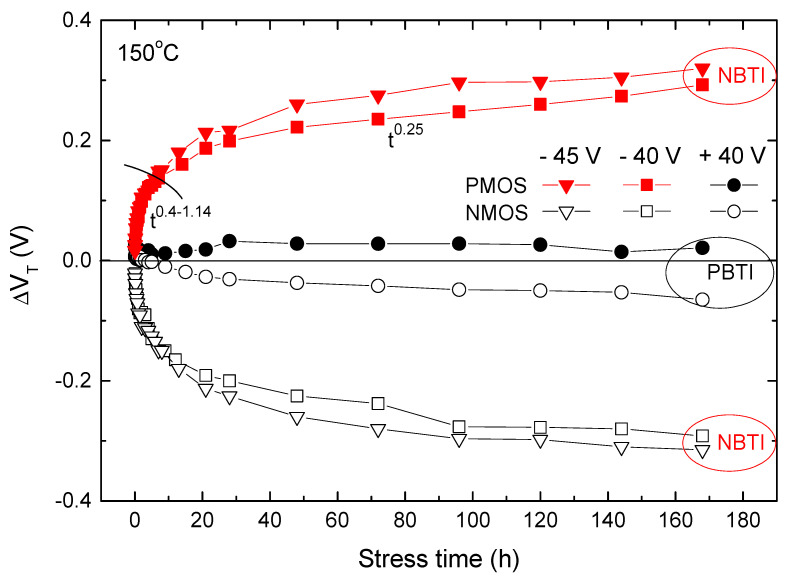
Threshold voltage shifts during continuous NBT and PBT stresses in NMOS and PMOS power VDMOSFETs.

**Figure 5 micromachines-15-00503-f005:**
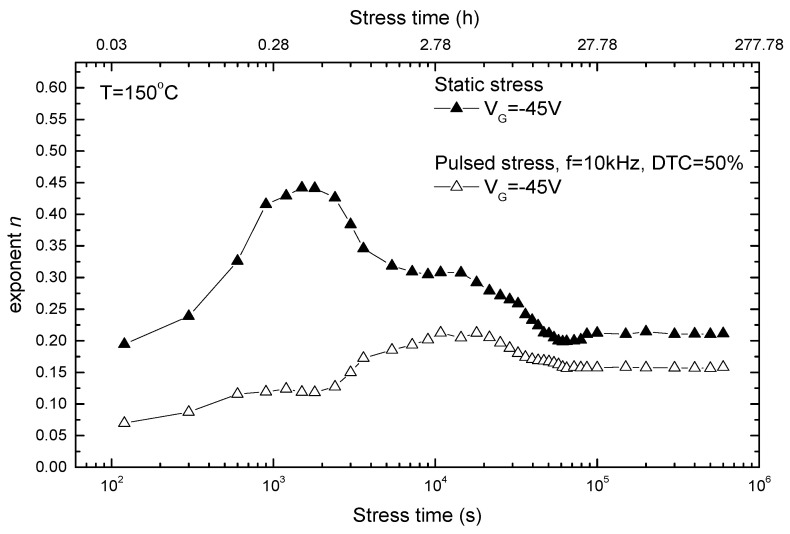
Variations of parameter *n* during the static and pulsed NBT stress.

**Figure 6 micromachines-15-00503-f006:**
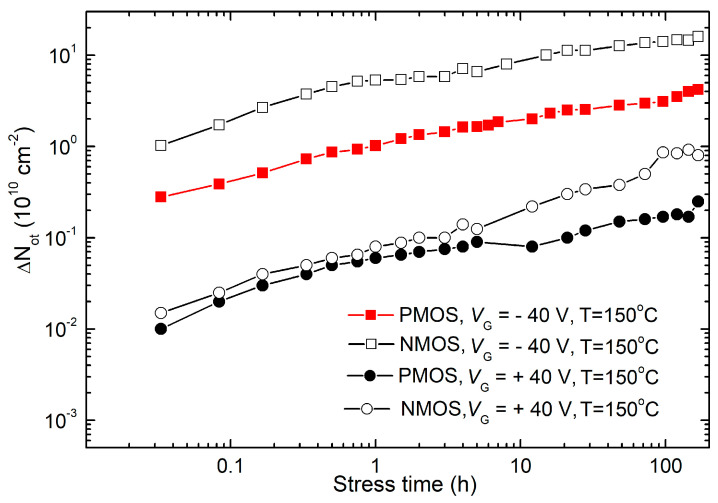
Time dependencies of Δ*N_ot_* during NBT and PBT stresses in both p- and n-channel VDMOS power transistors, determined via the SMG technique.

**Figure 7 micromachines-15-00503-f007:**
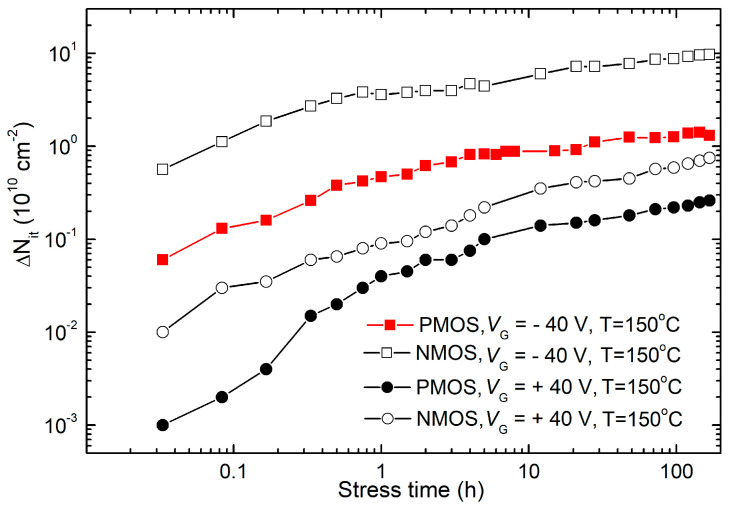
Time dependencies of and Δ*N_it_* during NBT and PBT stresses in both p- and n-channel VDMOS power transistors, determined via the SMG technique.

**Figure 8 micromachines-15-00503-f008:**
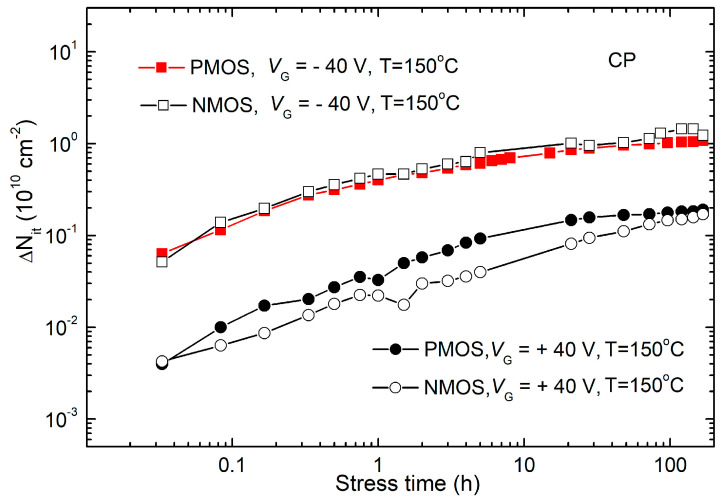
Time dependencies of and Δ*N_it_* during NBT and PBT stresses in both p- and n-channel VDMOS power transistors, determined via the CP technique.

**Figure 9 micromachines-15-00503-f009:**
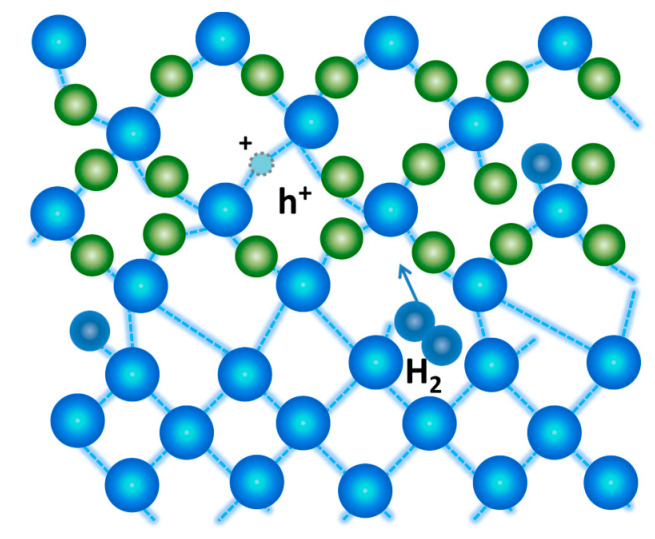
Breaking of weak ≡Si_O_-H bonds near the silicon-silicon dioxide interface.

**Figure 10 micromachines-15-00503-f010:**
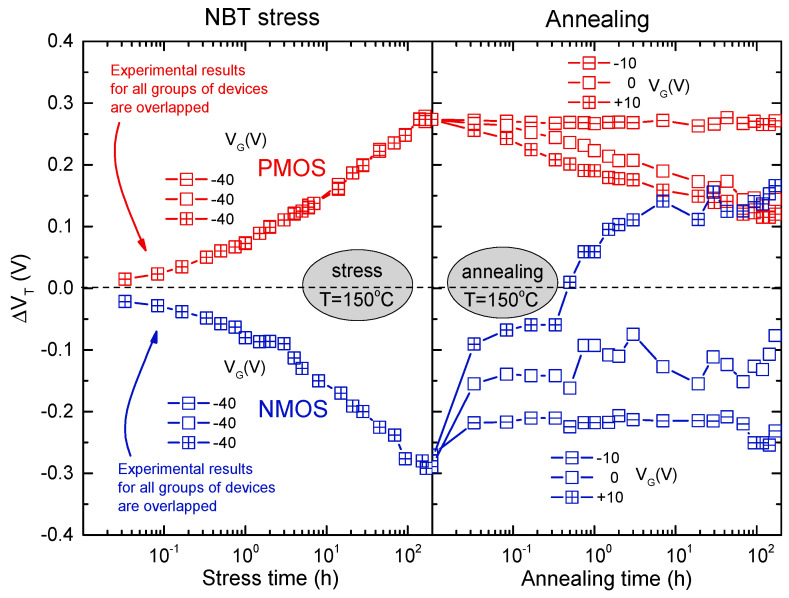
Threshold voltage shifts during NBT stressing and gate bias annealing of PMOS (in absolute value) and NMOS devices [12].

**Figure 11 micromachines-15-00503-f011:**
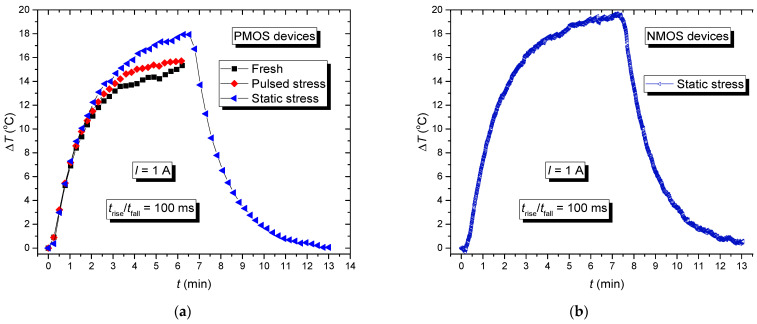
The changes of absolute temperature value, (**a**) PMOS devices with previous different types of stresses; (**b**) NMOS device with previous static NBT stress.

**Figure 12 micromachines-15-00503-f012:**
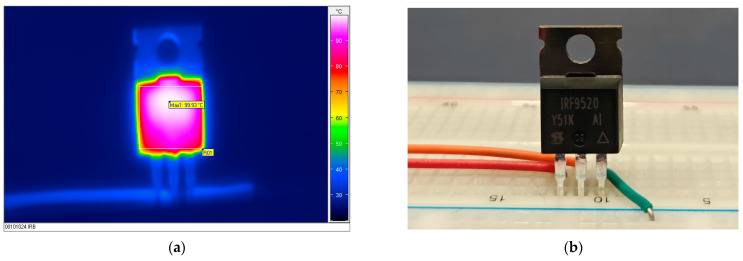
Infrared picture (**a**) and photo (**b**) of PMOS power transistor during the last part of the experiment.

**Table 1 micromachines-15-00503-t001:** The conditions under which NMOS and PMOS transistors can be subjected.

Polarization of the Gate of VDMOS Transistors	NMOS	PMOS
negative	NBTI (significant) For special purposes [12])	NBTI (significant) Normal operating conditions
positive	PBTI (insignificant) Normal operating conditions	PBTI (insignificant)

## Data Availability

The original contributions presented in the study are included in the article, further inquiries can be directed to the corresponding author.

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
