# Peer review of "A Reliability Investigation of VDMOS Transistors: Performance and Degradation Caused by Bias Temperature Stress"

_micromachines, 2024, doi:10.3390/mi15040503_

Round 1
Reviewer 1 Report
Comments and Suggestions for Authors
The work by Zivanovic et al is well written and well structured. The introduction states in a concise way the motivation of the present study and why it is relevant in the field of power VDMOSFETs. The used tools, the procedure setup and the range of used values are described in detail. The use of commercial p-channel (IRF9520) and n-channel (IRF510) power transistors for the study is appealing and useful. The results are explained in a concise and rigorous manner and the reasons behind the observed results are analysed in detail. In the conclusions, the authors explain the future steps and experiments.
Therefore, the article is ready to be accepted and published in the journal Micromachines.
I would just ask kindly to specify the meaning of the open and filled symbols in the graphs from the figures 6, 7 and 8, either in the captions or in the insets, in order to make it easier for the readers to understand when they look at the figures. I also kindly ask to improve the quality of the image from figure 9.
Comments on the Quality of English LanguageThe article is well written in general. The design of the experiments, the observed results after the experiments and the analysis of these results are well explained and easy to follow and understand.
Some grammatical details or the lack of some definite/indefinite articles in several sentences could be checked at the proof stage of the manuscript if it is accepted for publication.
Reviewer 2 Report
Comments and Suggestions for Authors
and degradation of both p- and n-channel Vertical Double Diffused MOS (VDMOS) transistors under bias temperature stress. Findings reveal meaningful threshold voltage shifts in both PMOS and NMOS devices due to stresses, and the subsequent annealing process was analyzed in detail. This work also examines the influence of stress history on self-heating behavior under real operating conditions. Additionally, the study elucidates the complex correlation between stress-induced degradation and device reliability. The insights gained contribute to optimizing the performance and permanence of VDMOS transistors in practical applications, advancing semiconductor technology. The manuscript is well-crafted, presenting the data in a clear and concise manner. Given the significance of these findings to the field of Micromachines, I believe the manuscript would be of great interest to the readers. Therefore, I recommend publication of this manuscript after addressing the following points:
1. Don't write any introduction in "abstract", but go right to the point. So eliminate this paragraph from Abstract: " Bias Temperature Instability (BTI) represents a significant challenge in advanced CMOS devices, particularly affecting p-channel Metal Oxide Semiconductor (PMOS) transistors. Although historically supposed to be less vulnerable, studies suggest potential vulnerability of n-channel Metal Oxide Semiconductor (NMOS) transistors under specific conditions".
2. In the introduction section, it is recommended that the authors provide more background information about NMOS, delineating their technological evolution and pivotal role in semiconductor technology. Such an elaboration will significantly bolster the reader's comprehension of the study's context, underlining its significance and situating it within the broader research landscape. (Ref: Science China Materials 2023, 66: 3251; Acs Nano 2023, 17:16912;)
3. In recent years, there has been significant research conducted both domestically and internationally on electrospinning technology and its applications in the field of field-effect transistors (FETs) based on metal oxide nanowires. Therefore, it is advisable for the authors to provide a brief introduction to the potential characteristics and applications of nanowire MOSFETs. (Ref: Nano Letter 2023, 23:7364; Advanced Fiber Materials 2023, 5:12; Science China Materials 2023, 66: 4445; Advanced Fiber Materials 2023, 5:1919-1933;)
4. ly encapsulate the import of the research findings. The authors should ensure that this section provides a thorough synthesis of the study’s results, emphasizing their relevance and contributions to the field.
5. It is kindly requested that the Figures in your paper, especially Figure 9, be replaced with clearer ones and that the axis format be corrected. The use of clearer and properly formatted images is recommended.
6. Please unify the format of references in the manuscript.
Comments on the Quality of English LanguageThe authors should get help from native English speaker to polish the manuscript.
Round 2
Reviewer 2 Report
Comments and Suggestions for Authors
The authors have replied all the questions and made relative revisions.